# Vibron-assisted spin excitation in a magnetically anisotropic molecule

N. Bachellier[1], B. Verlhac [1✉], L. Garnier[1], J. Zaldívar[2], C. Rubio-Verdú[2], P. Abufager[3], M. Ormaza [1,4], D.-J. Choi[5,6], M.-L. Bocquet [7], J.I. Pascual[2,8], N. Lorente[5,9] & L. Limot [1✉]

The electrical control and readout of molecular spin states are key for high-density storage. Expectations are that electrically-driven spin and vibrational excitations in a molecule should give rise to new conductance features in the presence of magnetic anisotropy, offering alternative routes to study and, ultimately, manipulate molecular magnetism. Here, we use inelastic electron tunneling spectroscopy to promote and detect the excited spin states of a prototypical molecule with magnetic anisotropy. We demonstrate the existence of a vibron-assisted spin excitation that can exceed in energy and in amplitude a simple excitation among spin states. This excitation, which can be quenched by structural changes in the magnetic molecule, is explained using first-principles calculations that include dynamical electronic correlations.

[1] Université de Strasbourg, CNRS, IPCMS, UMR 7504, F-67000 Strasbourg, France. [2] CIC nanoGUNE, 20018 Donostia-San Sebastián, Spain. [3] Instituto de Física de Rosario, Consejo Nacional de Investigaciones Científicas y Técnicas (CONICET) and Universidad Nacional de Rosario, Av. Pellegrini 250 (2000), Rosario, Argentina. [4] Universidad del País Vasco, Dpto. Física Aplicada I, 20018 Donostia-San Sebastián, Spain. [5] Centro de Física de Materiales (CFM MPC) CSIC-EHU, 20018 Donostia-San San Sebastián, Spain. [6] Ikerbasque, Basque Foundation for Science, Bilbao, Spain. [7] PASTEUR, Département de Chimie, Ecole Normale Supérieure, PSL University, Sorbonne Universités, CNRS, 24 Rue Lhomond, 75005 Paris, France. [8] Ikerbasque, Basque Foundation for Science, Bilbao, Spain. [9] Donostia International Physics Center (DIPC), 20018 Donostia-San Sebastián, Spain. ✉email: verlhac@ipcms.unistra.fr; limot@ipcms.unistra.fr

Electrically-driven excitations among spin states[1], or spin excitations, are increasingly observed in organometallic molecules coupled to metallic electrodes[2–10], and show great potential in view of manipulating the molecular spin[4,11]. In these molecules, the reduced symmetry of the metal center aligns the magnetic moment of the molecule along certain favorable directions and spin excitations can be produced in the absence of a magnetic field owing to this magnetic anisotropy[12]. Along with spin excitations, electrons may also trigger molecular vibrational modes, or vibrons, that modify[13–18] or even suppress[19,20] molecular conductance. Even if the presence of both excitations could be evidenced in a molecular system[21,22], experimental observations regarding their interplay have remained surprisingly elusive.

Individual molecular spins are also prone to the Kondo screening by the host electrons of the metal electrode. In this case, the electron–vibron interaction can produce resonances in the molecular conductance at the bias of the vibron's excitation energy[23–27]. These resonances are ascribed to tunneling electrons that have their spin flipped when elastically scattering off the molecular spin, but with sufficient energy to activate a vibrational mode in the molecule[28]. In principle, a similar scattering mechanism can be expected for tunneling electrons losing energy to both the molecular spin states and a vibrational mode.

With this in mind, here we use scanning tunneling microscopy (STM) to study a molecular complex with magnetic anisotropy, which includes a nickelocene molecule [$Ni(C_5H_5)_2$, see Fig. 1d; noted Nc hereafter] and a Ni atom. When the complex (noted NiNc hereafter) is embedded in a Nc layer on Cu(100), sizable spin and vibrational excitations can be electrically-driven in NiNc at distinct threshold energies. With the help of density functional theory (DFT) calculations, we demonstrate that a joint spin-vibration excitation is also present in NiNc resulting from the concomitant excitation of a spin and a vibration. This excitation should be quite common to magnetic molecules.

## Results

**Layer-integrated NiNc.** After deposition of Nc onto the Cu(100) surface (see Methods section), well-ordered molecular assemblies on the surface were found (Fig. 1a), along with isolated Nc molecules (Fig. 1b). The ring-shaped pattern in the images is produced by a cyclopentadienyl (Cp hereafter) ring and indicates that Nc is adsorbed with its principal axis perpendicular to the surface[29]. In the molecular layer, however, these "vertically" adsorbed molecules coexist with "horizontally" adsorbed molecules (principal axis parallel to the surface), as sketched in the inset of Fig. 1a. This T-shaped configuration is governed by van der Waals interactions[30] and results in two possible molecular configurations, known as paired (Fig. 1a) and compact (not shown)[29]. Our experimental observations regarding the formation and properties of the NiNc complex showed no significant difference between the two configurations, therefore in the following we will only focus on the paired configuration.

To build NiNc complexes, we proceeded as in earlier work on ferrocene[31] and exposed the Nc layer to single Ni atoms (Fig. 1a). The molecular complex is imaged as a ring with an apparent height of $5.8 \pm 0.2$ Å relative to the underlying copper surface (Fig. 1b, c), while the neighboring Nc molecules have an apparent height of $4.1 \pm 0.2$ Å. Similar to previous experiments in which cobalt was deposited onto a ferrocene layer[31], the ring-like shape demonstrates that the atom is positioned beneath a Nc molecule. This assignment is confirmed by our DFT calculations (Fig. 1d–f; details of the calculation are given in the Methods section] showing a 3 eV energy difference in favor of the deposited Ni atom beneath Nc rather than on top. The Ni atom,

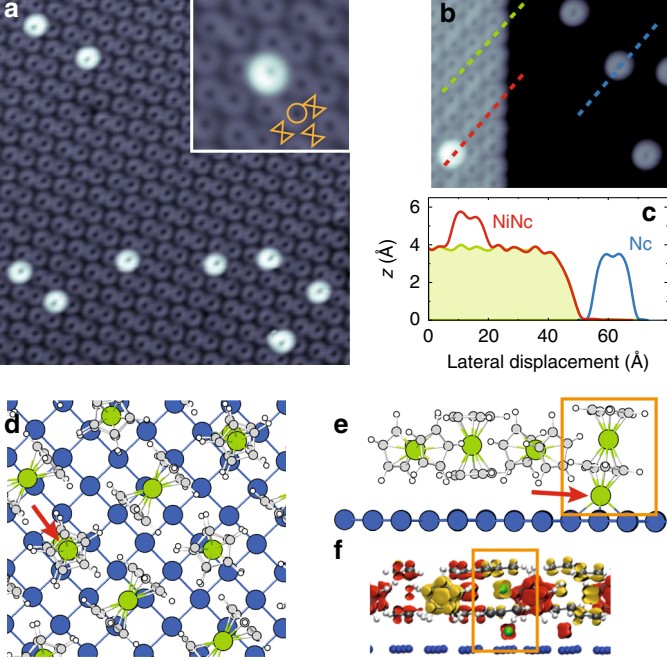

**Fig. 1 NiNc adsorption in a nickelocene layer. a** Self-assembled layer (paired configuration) and NiNc complex on Cu(100) (image size: $16 \times 16$ nm$^2$, sample bias: 20 mV, tunneling current: 20 pA). Inset: Close-up view of a NiNc complex in the paired layer. The molecular arrangement is sketched in orange as a circle for the vertical Nc and as a hourglass for the horizontal Nc ($4 \times 4$ nm$^2$, 20 mV, 100 pA). **b** Edge of a paired layer and isolated Nc molecules on Cu(100) ($10 \times 10$ nm$^2$, 20 mV, 100 pA) and **c** their corresponding height profiles along the dashed lines in **b**. **d** Top view, **e** side view of the DFT optimized structure of the NiNc complex embedded in a Nc paired layer—some molecules have been removed for clarity (H: white, C: gray, Ni: green). The red arrow indicates the position of the Ni adatom. **f** Side view of the spin density map of the paired structure containing two Ni atoms. Orange rectangles mark the NiNc complex. Yellow: Spin up, red: Spin down.

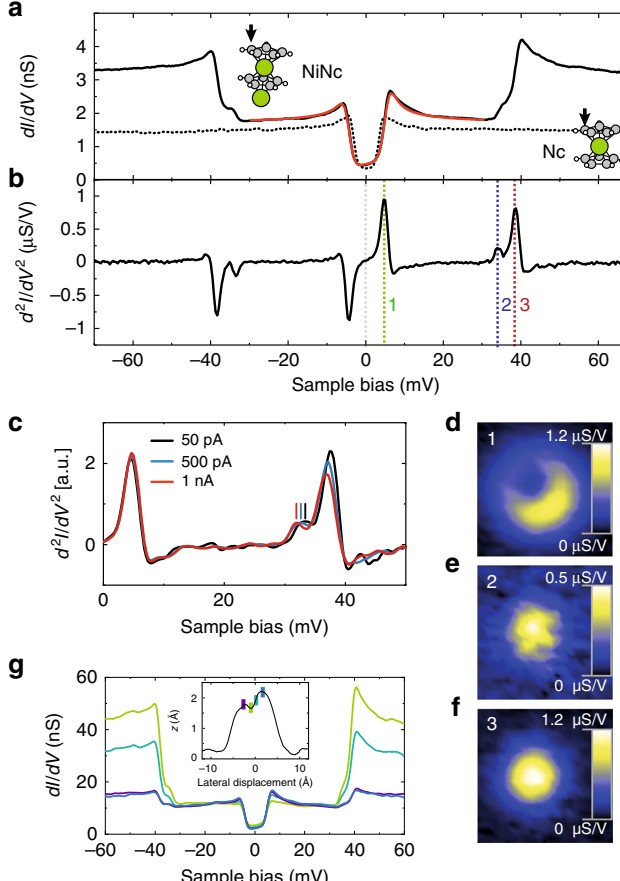

**Fig. 2 Local spectroscopy of NiNc. a** d$I$/d$V$ spectrum and **b** d$^2I$/d$V^2$ spectrum acquired above the Cp ring of a NiNc complex (feedback loop opened at −80 mV and 200 pA). The solid red line is a fit based on a dynamical scattering model[34]. The data exhibited a negligible tip dependence as reflected by the measurement uncertainties over a collection of NiNc complexes. The energy onsets are labeled **1**, **2**, and **3** in **b**. The dashed curve in **a** is the d$I$/d$V$ spectrum acquired above the Cp ring of a Nc in the layer (feedback loop opened at −70 mV and 100 pA). Inset: Arrow indicates the position where the spectrum was acquired. **c** d$^2I$/d$V^2$ spectra acquired above the center of a NiNc complex at various tip-molecule distances. The feedback loop was opened at 50 pA (solid black line), 500 pA (solid blue line) and 1 nA (solid red line) with a bias set to −20 mV—higher opening currents correspond to smaller tip-molecule distances. The spectra were normalized by the opening current. The vertical lines indicate the peak position of excitation **2**. **d**–**f** Spatial variation of the d$^2I$/d$V^2$ signal acquired at a constant-height and at a bias of 4.5 mV, 32.5 mV, and 37 mV, respectively (feedback loop opened at −60 mV and 500 pA above a Nc molecule of the layer; 3 mV rms modulation amplitude). These biases correspond to the excitation thresholds highlighted in **b**. **g** d$I$/d$V$ spectra acquired at four locations above NiNc [indicated by colored dots in the line profile of NiNc presented in the inset]. The feedback loop was opened at −30 mV (300 pA) above all locations in order to have the same amplitude for step **1** in the d$I$/d$V$ spectra.

which for clarity we refer to as Ni adatom hereafter, is located 2.4 Å above the copper surface. The NiNc complex displays a lower symmetry with a 0.5 Å misalignment between the two Ni atoms (Fig. 1d) and a tilt of the principal axis of Nc (Fig. 1e). This tilt is confirmed by close-up STM images [see inset of Fig. 1a and line profile in Fig. 1c], differentiating the present NiNc complex from those investigated numerically in previous studies, where

the Ni adatom is centered on the ring[32,33]. We show below that NiNc adopts instead this structure outside the molecular layer.

Figure 2a presents a typical d$I$/d$V$ spectrum acquired above the Cp ring of a NiNc complex in the paired layer, while the d$^2I$/d$V^2$ spectrum is shown in Fig. 2b. The d$I$/d$V$ spectrum is dominated by stepped features, symmetric with respect to zero bias, which point to inelastic excitations. The energy onset of these steps, as determined over a collection of NiNc complexes, are $|\epsilon_1| = 4.3 \pm 0.4$ meV (the excitation is noted **1** hereafter), $|\epsilon_2| = 33.9 \pm 0.5$ meV (noted **2**) and $|\epsilon_3| = 38.1 \pm 0.6$ meV (noted **3**). The data exhibited a negligible tip dependence.

**Spectroscopic assignment.** Given the similarity to the spin excitation spectrum measured above Nc [dashed line in Fig. 2a], which is known from previous studies[8,34], we assign **1** to a spin excitation. To confirm this assignment, we carried out DFT calculations (see Methods section). Using the relaxed structure of NiNc determined above, we find that the $d_{xz}$ and $d_{yz}$ frontier orbitals of Nc in the complex are spread out in a range of ±1 eV around the Fermi level (Supplementary Fig. 1). The NiNc complex has a total magnetic moment of 1.4 $\mu_B$, corresponding to an antiferromagnetic coupling between the Ni adatom (−0.2 $\mu_B$) and Nc (+1.6 $\mu_B$) with a charge transfer of 0.1 electrons from the substrate. The DFT calculations point therefore to an effective spin of $S = 1$. The strong resemblance with the spin excitation of single Nc molecules[8,35] further allows us to identify the spin spectrum with the one originating from a $S = 1$ system. Consequently, we model **1** via a spin Hamiltonian that includes axial magnetic anisotropy

$$\hat{H}_0 = \hat{H}_A + DS_z^2, \qquad (1)$$

where $\hat{H}_A$ is an Anderson Hamiltonian involving a single Nc orbital of the NiNc complex (see Methods section). The z-axis is chosen along the molecular axis running through the center of the Cp rings. Within this viewpoint, **1** is assigned to a spin excitation occurring between the ground state $|S = 1, M = 0\rangle$ and the doubly degenerate $|S = 1, M = \pm 1\rangle$ excited states of NiNc [see Fig. 2a], the onset $|\epsilon_1|$ corresponding then to $D$. The fit to the line shape based on Eq. (1) is highly satisfactory [solid red line in Fig. 2a][34], and yields $D = 4.6 \pm 0.2$ meV. The inclusion of many-body interactions in the fit through $\hat{H}_A$, i.e., the inclusion of Kondo-like phenomena, is crucial for reproducing the cusp observed above the energy threshold of the spin excitation[9,36,37]. The cusp is associated to a Kondo fitting parameter $\mathcal{J}\rho = -0.4 \pm 0.2$ that is typical for Nc on copper substrates (Supplementary Fig. 2 and Supplementary Note 1).

Excitation **2** corresponds instead to a vibrational excitation of energy $|\epsilon_2| = \hbar\omega = 33.9 \pm 0.5$ meV. When approaching the tip towards NiNc, we observe a red shift of the peak (dip) associated to **2** as high as 1.5 meV (Fig. 2c and Supplementary Table 1), which is characteristic of a vibrational excitation[38,39]—the onset of **1** is instead nearly constant. The spatial dependence of the d$^2I$/d$V^2$ signal above the NiNc molecule also hints to a vibrational excitation. While excitation **1** is located on the Cp ring (Fig. 2d), excitation **2** is instead maximal in the center of the ring (Fig. 2e). To elucidate this difference, we computed the vibrational modes for the relaxed NiNc structure determined above. We found three vibrational modes at 29.1, 31.8 meV (Fig. 3a), and 35.5 meV (Fig. 3b) close to the experimental energy $|\epsilon_2|$, all of them being robust to the finite atomic displacements used in the calculation. The first mode corresponds to a molecular frustrated rotation, which we discard as is at variance with the experimentally observed spatial location of the vibration. The second and third mode correspond to a translational motion of the Ni atom inside Nc. The tilted adsorption geometry of Nc on the Ni adatom breaks the degeneracy of these two modes observed in the gas

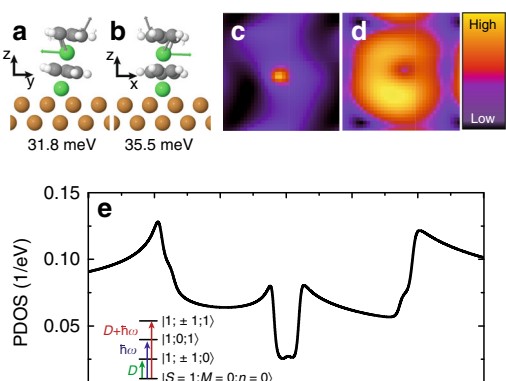

**Fig. 3 Active vibrational modes and simulated spectrum of NiNc. a, b** Calculated vibrational modes close to $|\epsilon_2|$. **c** Computed ratio between the inelastic and elastic tunneling conductance using the 35.5 meV mode. The maximum change represents 5% of the elastic conductance and is located in the center of the molecule. **d** Calculated local density of states of a NiNc complex at the Fermi level. **e** Projected density of states (PDOS) on the occupied molecular orbital as a function of electron energy calculated from Eq. (2). In the tunneling regime, the differential conductance is proportional to the PDOS[42]. Inset: State diagram of a NiNc complex based on Eq. (2); the eigenstates are noted as $|S, M; n\rangle$ and the arrows depict the three transitions leading to the excitation steps in the d$I$/d$V$ spectrum.

phase. While the 31.8 meV mode only gives a negligible change in the simulated conductance across the molecule, we find instead a dominant contribution for the 35.5 meV mode with a spatial dependence matching experimental observations (Fig. 3c). The simulated STM image of NiNc is presented in Fig. 3d[40]. Excitation **2** is then assigned to a Ni-Cp mode (Fig. 3b), where the internal Ni atom moves parallel to the tilted Cp ring.

Excitation **3** shares spectroscopic fingerprints with excitations **1** and **2**: (i) all the spectra recorded so far showed that the energy onset of **3** is the sum of the energy onsets of **1** and of **2**, $|\epsilon_3| = |\epsilon_1| + |\epsilon_2|$ (Supplementary Table 2); (ii) the line shape of **1** and **3** exhibit a cusp above their corresponding excitation energies, which, as stressed above, is a characteristic feature of a spin excitation; (iii) a red shift is observed for **3** when approaching the tip towards NiNc as seen for excitation **2** (Fig. 2c); (iv) **2** and **3** have same spatial distribution over the NiNc complex (d$^2I$/d$V^2$ maps of Fig. 2e, f), their step amplitudes in the d$I$/d$V$ spectra varying proportionally to one another across the NiNc molecule with a ratio of $3 \pm 1$ (Fig. 2g). These results are remarkable and raise the question of how a second spin excitation can be present in the NiNc complex and how it relates to a vibrational excitation.

**Model Hamiltonian and electronic transport.** In order to model the experimentally observed d$I$/d$V$ spectrum, we extend the spin Hamiltonian of Eq. (1) to include vibrational effects

$$\hat{H} = \hat{H}_0 + \hbar\omega(\hat{b}^\dagger\hat{b} + \frac{1}{2}) + W \sum_\sigma \hat{d}^{\sigma\dagger}\hat{d}^\sigma(\hat{b} + \hat{b}^\dagger). \quad (2)$$

As previously, we take a single $d$-level for the molecule and introduce the states $d$ annihilated and created by $\hat{d}^\sigma$ and $\hat{d}^{\sigma\dagger}$ with spin $\sigma$, respectively. We also assume that a molecular vibration of frequency $\omega$ is annihilated and created by $\hat{b}$ and $\hat{b}^\dagger$, respectively, and, moreover, that the electron and vibration couple with a strength $W$ when the state $d$ is populated. We use a weak electron-vibration coupling of $W = 20$ meV as estimated from the DFT

calculations presented above[41]. Equation (2), which does not include a spin-vibron coupling, shows that a vibron is excited when an electron tunnels into the molecule. This in turn affects the electronic correlation included in $\hat{H}_A$, as well as the probability of spin excitation, which strongly depends on the occupation of the molecular orbital[37]. Figure 3e presents the computed spectrum based on Eq. (2) (see Methods section)[42], where $D = 4.6$ meV and $\hbar\omega = 35.5$ meV. The computed spectrum showed little dependence on the parameters used for the molecular orbital. The electron–phonon interaction in the presence of spin excitations is sufficient to reproduce the experimental spectrum, but the inclusion of dynamical electronic correlations in the calculation is essential to correctly grasp the amplitude of the vibrational step.

The inset of Fig. 3e sketches the corresponding eigenenergies and allowed excitations. The first and second excited states correspond to a spin excitation $|S = 1, M = \pm 1; n = 0\rangle$ and to a vibrational excitation $|S = 1, M = 0; n = 1\rangle$, respectively. The third excited state, $|S = 1, M = \pm 1; n = 1\rangle$, corresponds to a spin excitation energetically displaced upward in energy by a vibron. This excitation mechanism is similar to the coupled-spin vibrational Kondo effect and its replicas of the Kondo resonance in the tunneling spectra at energies close to $\hbar\omega$. Consistent with this assignment, the relation between the three excitation thresholds simply reflects $|\epsilon_3| \simeq D + \hbar\omega$. The distortion of the NiNc molecule is negligible when the molecular vibration is active as expected for a weak electron-vibration coupling. A renormalized value of $D$ would be observed otherwise leading to $|\epsilon_3| \neq |\epsilon_1| + |\epsilon_2|$[43,44]. The relative amplitudes of the steps in the d$I$/d$V$ spectrum of NiNc can also be explained using this framework. Noting the step amplitudes by $\sigma_1$, $\sigma_2$, and $\sigma_3$ (Supplementary Fig. 3 and Supplementary Note 2), we find that all spectra recorded obey $\sigma_3/\sigma_0 = (\sigma_2/\sigma_0)(\sigma_1/\sigma_0)$ (Supplementary Table 3), where the vacuum barrier thickness is accounted for by the elastic contribution $\sigma_0$. This relation indicates that the spin and the vibrational excitations occur independently one from another with transition rates proportional to $\sigma_1/\sigma_0$ and $\sigma_2/\sigma_0$, respectively, while the transition rate of the combined excitation $\sigma_3/\sigma_0$ is their product. This can lead, eventually, to a vibron-assisted spin excitation that exceeds in intensity the spin excitation (Fig. 2g). The same relation among step amplitudes was observed for single and double spin excitations produced by one electron tunneling across two magnetic molecules[8].

**Isolated NiNc.** For completeness, we highlight the importance of the Nc layer for observing the inelastic excitations in the NiNc complex. For this purpose, we engineered the NiNc complex outside the layer via a tip-assisted manipulation[31]. To do so, we first transferred an isolated Nc to the tip[8,35] and then transferred it back atop an isolated Ni adatom on the surface (Fig. 4a, b). The newly formed molecule (Fig. 4b) has an apparent height of $5.6 \pm 0.2$ Å exceeding that of an isolated Nc molecule, $3.5 \pm 0.2$ Å (Fig. 4c), and presents a perfectly ring-shaped pattern, indicating that this complex is not tilted but lies straight. The d$I$/d$V$ spectrum changes completely compared to NiNc in the layer, showing now a broad resonance centered near the Fermi level (Fig. 4d); no inelastic excitation could be evidenced. DFT calculations give a clear picture of the new adsorption configuration adopted by NiNc. The Ni adatom, which is adsorbed in a hollow position of Cu(100), is now centered on the Cp ring (Fig. 4e) and located at a distance of 2.47 Å from the copper surface. This higher adsorption symmetry for NiNc follows from the absence of steric constraints with neighboring Nc. The lowest unoccupied molecular orbital of an isolated NiNc can be qualitatively represented by the mixing of the $p_z$ orbitals of C and of the $d_{xz}$ and $d_{yz}$ orbitals of Ni placed around the Fermi level resulting in a calculated magnetic moment of 1.17 $\mu_B$. The resonance of Fig. 4c is well described by a Frota-Fano fit[45] and is then assigned to a spin-1/2

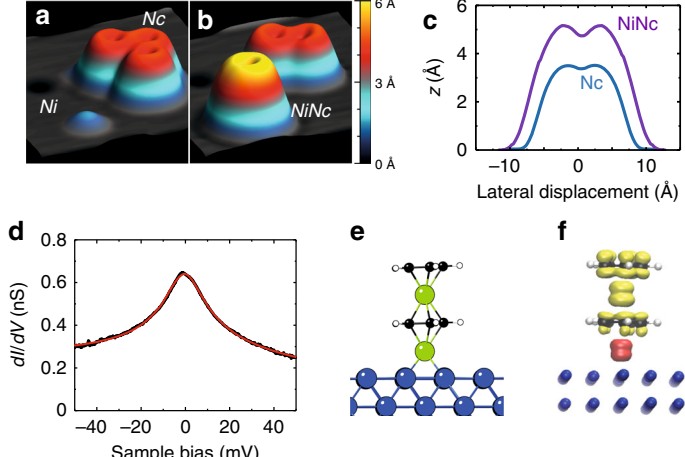

**Fig. 4 Isolated NiNc on Cu(100). a, b** Pseudo 3D images showing the tip-assisted assembly of an isolated NiNc complex ($5 \times 5$ nm$^2$, $-20$ mV, 20 pA): **a** before and **b** after the transfer of Nc atop an isolated Ni adatom, and **c** height profiles of isolated NiNc and of isolated Nc. **d** d$I$/d$V$ spectrum measured above the Cp ring of an isolated NiNc on Cu(100) (feedback loop opened at $-50$ mV and 20 nA). The solid red line is a Frota-Fano fit yielding a resonance centered at $-0.2 \pm 0.2$ meV and $T_K = 68 \pm 7$ K. **e** DFT optimized structure of isolated NiNc on Cu(100), and **f** Side view of the spin density map.

Kondo effect, the fit yielding a Kondo temperature of $T_K = 68 \pm 7$ K. The Kondo effect is carried by the $d_{xz}$ and $d_{yz}$ frontier orbitals of Nc as 94% of the spin density of NiNc is located on Nc. Compared to the NiNc complex of the layer, the Ni adatom likely provides a hybridization pathway to the copper surface due to its central adsorption in the Cp ring, causing the effective spin of NiNc to drop, but simultaniously promoting the Kondo physics[35]. The change of symmetry for the isolated NiNc compared to layer-integrated NiNc may also explain the absence of vibrational signature in Fig. 4d.

## Discussion

To summarize, we have shown that Nc adsorbed on a Ni atom yields a vibron-assisted spin excitation at an energy that is one order of magnitude higher than usual spin-excitation energies. We have demonstrated the general character of this excitation through a model that includes magnetic anisotropy, intramolecular correlations and electron-phonon coupling. Our findings suggest that the vibron-assisted spin excitation can be present in molecules with magnetic anisotropy and that it is easier to detect than the vibrational excitation associated to it. With this in mind, we assign the excitation detected above the center of layer-integrated Nc molecules (Supplementary Fig. 4) to a vibron-assisted spin excitation.

While preparing the revised manuscript, a related study to the present one was published by another group[46].

## Methods

**Experimental details**. The measurements were performed in an ultra-high vacuum STM operating at 2.4 K. The Cu(100) surface was cleaned in vacuo by sputter/anneal cycles, while a sputter-cleaned etched tungsten tip was employed for tunneling. The tip was further prepared by controlled tip-surface contacts to ensure a mono-atomically sharp copper apex. All the spectra were recorded with a lock-in amplifier (200 µV rms and 716 Hz) using a tip that was verified to have a negligible electronic structure in the bias range investigated. The d$^2$I/dV$^2$ spectra were numerically derived. Nickelocene was deposited onto the cold (<100 K) Cu(100) surface. A molecular flux of $2.5 \times 10^{-2}$ monolayer/min was used in order to obtain well-ordered molecular assemblies along with isolated Nc molecules. To build NiNc complexes, we then exposed the surface to a small amount of Ni atoms (0.05 monolayers). The single-nickel atoms were deposited from a Ni wire source (99.99% purity) onto the cold surface ($\approx 10$ K) through an opening in the cryostat shields.

**Computational details**. We have continued the work of previous publications[8,35] treating the adsorption, chemical and physical properties of adsorbed Nc molecules on Cu(100). In the present case, a Ni atom was added to the substrate and the adsorption of Nc on this nucleation center was studied. Electron transmission calculations were also performed at the density functional theory (DFT) level. In this

way, two implementations of DFT were used: (i) VASP[47–52] for the adsorption, (ii) TRANSIESTA for the transport calculations[53]. The molecular geometry was optimized using DFT at the spin-polarized generalized gradient approximation (GGA-PBE) level, as implemented in VASP[47–52]. In order to introduce long-range dispersion corrections, we employed the so-called DFT-D2 approach proposed by Grimme[54]. We used a plane wave basis set and the projected augmented wave (PAW) method with an energy cutoff of 400 eV. The two surfaces representing substrate and tip were modeled using a slab geometry with a $4 \times 4$ surface unit cell and six layers for the surface holding the tip-apex and the molecule and five layers for the approaching surface electrode. The k-point sampling was converged at $3 \times 3$, although the sampling was $11 \times 11$ for the transmission calculations. For these last calculations, The valence-electron wave functions were double-$\zeta$ plus polarization (DZP) basis sets for Nc and diffuse orbitals were used to improve the surface electronic description and a single-$\zeta$ plus polarization (SZP) basis set for the copper electrodes. The use of a DZP basis set to describe the adsorbate states is mandatory in order to yield correct transmission functions[55,56]. The vibrational modes of NiNc were calculated by diagonalizing the dynamical matrix obtained from VASP while only taking the degrees of freedom of the Nc molecule into account.

**Simulations**. The electronic structure is modeled by one single molecular orbital. In order to model the experimentally observed d$I$/d$V$ spectrum, we then write an Anderson Hamiltonian for a $S = 1$ molecular system with a single molecular orbital in the presence of a molecular vibration of frequency $\omega$ and axial magnetic anisotropy, $D$

$$\hat{H} = \hat{H}_A + \hbar\omega(\hat{b}^\dagger \hat{b} + \tfrac{1}{2}) + \sum_\sigma W \hat{d}^{\sigma\dagger} \hat{d}^\sigma (\hat{b} + \hat{b}^\dagger) + D(\hat{d}^{\uparrow\dagger}\hat{d}^\uparrow - \hat{d}^{\downarrow\dagger}\hat{d}^\downarrow + \hat{S}_{2z})^2. \quad (3)$$

where

$$\hat{H}_A = \sum_{k,\sigma} \epsilon_k \hat{c}_k^{\sigma\dagger} \hat{c}_k^\sigma + \sum_\sigma \epsilon_d \hat{d}^{\sigma\dagger} \hat{d}^\sigma + \sum_{k,\sigma} V_k^\sigma \hat{c}_k^{\sigma\dagger} \hat{d}^\sigma + \sum_\sigma U \hat{n}_d^\sigma \hat{n}_d^{\bar{\sigma}}. \quad (4)$$

$k$ represents the quantum numbers for the metal states, assumed to be one-electron ($\epsilon_k$ are the spinless bands and $\hat{c}_k^\sigma$ the corresponding annihilation operator for that one-electron state). For the molecule we take a single level $\epsilon_d$ and the states $d$ destroyed and created by $\hat{d}^\sigma$ and $\hat{d}^{\sigma\dagger}$ with spin $\sigma$ respectively. The molecular vibration is local and generated or destroyed by $\hat{b}^\dagger$ or $\hat{b}$. And the electron and vibration couple with a strength $W$ when the state $d$ is populated. Also the electron and spin are coupled via the axial anisotropy term $D\hat{S}_z^2 = (\hat{S}_{1z} + \hat{S}_{2z})^2$ that considers the total spin of the molecule projected on the $z$ axis. We assume that there is already one spin on the molecule, $S_{2z}$, and that the fluctuations of charge in the molecular level leads to changes in $\hat{S}_{1z} = \hat{d}^{\uparrow\dagger}\hat{d}^\uparrow - \hat{d}^{\downarrow\dagger}\hat{d}^\downarrow$.

By solving the above Hamiltonian, we can realistically approach the experimental situation where inelastic spin-flip transitions affect the electron transmission through the molecule. In order to do this, we use the multi-orbital non-crossing approximation (MONCA) in the implementation by Korytár and Lorente[37,57] that has been shown to reproduce inelastic spectra correctly[37]. Vibrations can be included using the self-consistency loops of MONCA as explained in the works by Roura-Bas and collaborators[58,59]. The vibron is excited when an electron jumps into the molecule during the transport process. This is strongly affected by the electronic correlation included in the Anderson Hamiltonian. But it also affects the probability of spin excitation, that strongly depends on the occupation of the molecular orbital[37].

## Data availability

The data that support the findings of this study are available from the corresponding authors upon reasonable request.

## Code availability

The code used for the calculations of this study is available from M.-L.B. and N.L. upon reasonable request.

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

## Acknowledgements

We thank M. Ternes for providing his fitting program. This work was supported by the Agence Nationale de la Recherche (grants No. ANR-13-BS10-0016, ANR-11-LABX-0058 NIE and ANR-10-LABX-0026 CSC) and by the Agencia Española de Investigación (grants Nos. MAT2016-78293-C6-1-R and MDM-2016-0618). D.-J.C. and N.L. thank the MICINN (project RTI2018-097895-B-C44). M.-L.B. thanks the national computational center CINES and TGCC (GENCI project: A0030807364).

## Author contributions

L.L. conceived the experiments. N.B., B.V., L.G., J.Z., M.O., D.-J.C., and L.L. performed the STM measurements. B.V., C.R.-V. analyzed and fitted the data. P.A., M.-L.B., and N.L. performed first principles density functional theory calculations. P.A. calculated the vibrational modes. N.L. developed the transport theory and simulated the differential conductance. B.V., J.I.P., N.L., and L.L. drafted the manuscript; all authors discussed the results and contributed to the manuscript.

## Competing interests

The authors declare no competing interests.
