## [Peer Review File · Nature Communications]

Reviewers' Comments:

Reviewer #1:

Remarks to the Author:

The authors showed that a molecular vibrational mode can be used to induce spin excitations of magnetic molecules. Combining scanning tunneling microscopy and DFT calculations, they studied the electronic and magnetic properties of a nickel-nickelocene complex (NiNc) on a metal surface, and found that the molecular complex embedded in a molecular layer of nickelocene has a vibrationally-assisted spin excitation at an energy much higher than the usual pure spin excitations. Overall, I think the authors present a new way to control spins in molecular magnets, which will be of interest to researchers in nanoscale magnetism and molecular spintronics. However, the following issues should be addressed before the publication.

1. The vibrational mode proposed for the layer-integrated NiNc should also be available to the isolated NiNc on Cu(001). Why is the vibrational mode not observed in the isolated NiNc on Cu as satellite Kondo peaks?
2. Is the dI/dV spectrum of Nc in Fig. 2a measured on an isolated Nc on Cu, or is it measured on an Nc in the molecular layer? This should be stated in the caption. The authors said additional dI/dV spectra are presented in Fig. S5 in the caption of Fig. 2. But Fig. S5 is missing. Any vibrational modes visible on the Nc in the layer? If not, what makes them visible in the NiNc in the molecular layer?
3. I am wondering if the authors have some ideas on why the other two vibrational modes (29.1 and 31.8 meV) as predicted by DFT are not detected from the point of view of selection rules.
4. The excitations 2 and 3 may come from the spin excitations of two coupled Ni atoms in the NiNc complex. How to rule out this possibility?
5. Have the authors studied the tip-height dependence of the dI/dV spectra of layer-integrated NiNc? Since the tip would affect the molecular vibrations, a shift of the vibrational energy or the change of the inelastic conductance is expected. These observations should be helpful to confirm the vibrational origin of the dI/dV features.
6. To distinguish the spin and vibrational features in the dI/dV , the authors may want to measure the dI/dV with a nickelocene-terminated tip. Only the spin excitation energy should split or shift with the presence of a magnetic tip. This is just an option.

Other minor points:

1. Experimental details (for example, how they cleaned the Cu substrate, dosed molecules and Ni atoms, lock-in parameters) should be moved to the Method section. I found them distracting when reading the paper.
2. Ref. 44 (Nano letters, 16, 588) from the same group already reported the method of producing metal-metalloocene complex, and thus should be cited when describing how the NiNc complex was built in the current manuscript.
3. Figs. 3a and 3b. Scale bars and color scales are missing; Fig. 3b. The energy of the density of

states should be stated.

Reviewer #2:

Remarks to the Author:

Review of "Vibron-assisted spin excitation in a magnetically anisotropic molecule"

N. Bachellier et al., have studied NiNc organometallic molecules using STM/STS. The authors have observed spin and vibrational excitation and the sum of the two excitations. Recent studies of Nc and NiNc complexes from the same group are definitely interesting in exploring molecular spin systems. However, I do not think that this work meets the high standard of novelty required by Nature Communications.

1) There have been numerous published results on measuring spin, vibrational, vibronic, spin-vibrational, Kondo-vibrational, and Kondo-spin IETS excitations of single atoms and molecules using STM/STS. In particular, there are two important works which already have reported spin and vibrational excitation. For example, Q. Dubout et al (PRL 114, 106807 (2015)) have performed STM measurements on individual Co atoms on Pt(111). Depending on chemical adsorption of H atom to Co atom, Q. Dubout et al., have not only measured spin anisotropy, Kondo, and vibrational IETS but also have manipulated the spin states of the Co atoms by controllably detaching H atom one by one. Furthermore, Q. Dubout et al., have distinguished the spin features vs vibrational features by measuring the shift of the IETS step with applying external magnetic fields.

Furthermore, Y.S. Fu et al (PRL 103, 257202 (2009)) have studied multi-layered CoPc molecules on Pb(111) surface using STM. Y.S. Fu et al., have measured spin-IETS as well as vibronic features and have utilized the change of the spin-IETS for identifying a charge state of molecules.

I think that previous researches in this field already cover the most of this work done by N.

Bachellier. Also, compared to the two papers mentioned above, I found that this work does not display enough controllability of the spins in the content of nanoscale spintronics.

2) The authors claim that a molecule shows the IETS step at the sum of the two excitations (spin-IETS and vibrational IETS). However, I found that the authors' claim is not convincing to me. For example, what if the IETS step of #2 is spin feature localized at the center of the complex? Could the authors perform magnetic field dependence measurement?

Even though the authors' claim is correct, there have been already many published results showing vibration assisted spin (Kondo) excitations. For example, T. Choi et al (Nano Lett 10, 4175 (2010)) have shown vibration assisted Kondo excitations which can be switched on/off reversibly by controlling the conformation of molecule.

3) The authors have assumed the complex has a uniaxial anisotropy as shown in eq. 1. However, the authors also have mentioned the complex has a canted geometry from Fig. 1a and Fig 2c. I believe the canted geometry may introduce in-plane magnetic anisotropy. So, I think the authors need to include in-plane anisotropy in eq. 1 and calculate the IETS energies as well as amplitude of the tunneling conductance at those IETS steps (C.F. Hirjibehedin et al., Science 319, 1066 (2007)).

4) In Fig. 3e, the authors have modeled that the IETS #3 is from vibrational mode ($n=1$) assisted spin-IETS. Have you observed the excitation at the energies of n being higher than 1 (ex, $n=2,3,\dots$) (For example, vibronic: X.H. Qiu et al., PRL 92, 206102 (2004), N.A. Pradhan et al., J. Phys. Chem. B 109, 8513 (2005), and spin+vibronic: Y.S. Fu et al (PRL 103, 257202 (2009))). In addition, why does the dI/dV map imaged at the IETS #3 (figure 2e) only show the localized signal at the center of complex? As the authors mentioned, the main contribution of IETS #3 is originated from magnetic anisotropy (IETS #1). However, the signal of the IETS #1 is localized at the ring of the complex not the center of the complex. Could the authors explain this discrepancy?

Because of the reasons listed above, I cannot recommend this manuscript to be published in Nature Communications.

Reviewer #3:

Remarks to the Author:

Bachelier and co-workers investigate complexes of a Ni atom with a nickelocene molecule. The authors find that in all cases the Ni atom sits below one of the Cp rings of the nickelocene molecule and provides a bond to the underlying copper surface. Interestingly, the authors find two very different adsorption geometries which lead to drastically different spectroscopic signatures in tunneling spectroscopy. First, when the complex is isolated from neighbors, the authors observe a Kondo effect with a surprisingly high Kondo temperature of 68K. Second, when the nickelocene complex is part of a network of molecules, the bonding geometry shows a lower symmetry due to van-der-Waals bonding to neighbors. This results in a visible tilt of the molecule and to a complete absence of the Kondo effect. Rather the spectra indicate a spin excitation, a vibrational excitation, as well as the combination of the two: a spin-vibration excitation at the sum of the two energies. Many results are convincingly confirmed and illustrated through spin-polarized density functional theory modeling.

This work is original, the manuscript is well written and the content of the spin properties of this class of molecule is a hot research topic at present. I am strongly in favor of publishing with minor modifications.

- 1) Page 1, line 42: I would suggest the authors distinguish the NiNc complex in the Nc network from the stand-alone NiNc because the vibrational mode is induced in the Nc network, is my understanding correct? The sentence can lead to misunderstandings as if the sizable vibrational mode occurred in NiNc itself.
- 2) Page 1, line 44 "previous excitations" is neither defined nor clear from context.
- 3) In Figure 2(c) and page 2, line 62, the apparent height of Nc is given as 4.1Å, while it is given as 3.5Å in page 5, line 149. Does the isolated molecule have a different apparent height compared to the one in the Nc layer? In Figure 2(c), the layer (green) and the isolated molecule (blue) look almost the same. If there is tip-dependence, the error given as 0.1Å seems too low.
- 4) In Figure 2, image scale for (c-e) is missing. Same for Figure 3 (a,b).
- 5) In Figure 2(a), the authors measured the two spectra at a different set point. Is there a special reason for this? If it is on purpose (to keep the sample-molecule distance same or other reason?), it would be helpful to mention it.
- 6) Page 3, line 85: the magnetic moment of the Ni atom and the nickelocene molecule are not close to integer or half-integer values. Yet the authors state without comment that the total spin is $S=1$. This seems to fall from the sky and should be discussed with more care.
- 7) Page 3, line 90: the value D is different from ϵ_1 . What is the relationship between the two values? If it is not trivial, it would be helpful for readers to mention or cite it.
- 8) Page 4, line 108: the sentence 'Second, 3 bears spectroscopic...' is not clear and easy to understand. Could the authors clarify this sentence?
- 9) Page 4, line 111: $dI_2/dV_2 \diamond d^2I/dV_2$
- 10) Figure caption of figure 4: I would state the FWHM of the curve – or the Kondo temperature, to help the reader. In Fig 4(c), the x-axis may be in nm. And, the caption for (d) is labeled as (c).

We are pleased to resubmit a revised version of our manuscript. Changes to the manuscript are highlighted in red. We thank the reviewers for their comments and criticism and hope that the modifications meet their expectations. Please note that the line numbering and references used in our answers refer to the new version of the manuscript and of the supplementary materials.

Response to Reviewer #1

The authors showed that a molecular vibrational mode can be used to induce spin excitations of magnetic molecules. Combining scanning tunneling microscopy and DFT calculations, they studied the electronic and magnetic properties of a nickel-nickelocene complex (NiNc) on a metal surface, and found that the molecular complex embedded in a molecular layer of nickelocene has a vibrationally-assisted spin excitation at an energy much higher than the usual pure spin excitations. Overall, I think the authors present a new way to control spins in molecular magnets, which will be of interest to researchers in nanoscale magnetism and molecular spintronics. However, the following issues should be addressed before the publication.

We thank the Reviewer for the positive assessment of our work. He/she is convinced by our findings and recommends publication, but raises some points that we carefully address here below.

(1) The vibrational mode proposed for the layer-integrated NiNc should also be available to the isolated NiNc on Cu(001). Why is the vibrational mode not observed in the isolated NiNc on Cu as satellite Kondo peaks?

The Reviewer raises an interesting point. However, he/she should keep in mind that the two NiNc complexes have a different chemical structure and their close environment differs strongly. In the layer-integrated NiNc, the two Ni atoms are misaligned by 0.5 Å [line 72 in the main text and Fig. 1(e)], while in the isolated NiNc the two Ni atoms are perfectly aligned [line 169 and Fig. 4(e)]. These intrinsic and extrinsic structural differences are strong and, *a priori*, the vibrational spectra should differ. In answer 2, for instance, we consider the simple case of a single nickelocene molecule (no complex). As we discuss, a change in its extrinsic structure is already sufficient to suppress the vibrational signature.

We have added lines 178-179 to account for the Reviewer's remark.

(2) Is the dI/dV spectrum of Nc in Fig. 2a measured on an isolated Nc on Cu, or is it measured on an Nc in the molecular layer? This should be stated in the caption. The authors said additional dI/dV spectra are presented in Fig. S5 in the caption of Fig. 2. But Fig. S5 is missing. Any vibrational modes visible on the Nc in the layer? If not, what makes them visible in the NiNc in the molecular layer?

Figure 2a presents a dI/dV spectrum of a layer-integrated Nc. This is now explicitly stated in the new version of the manuscript. The spectrum was acquired by positioning the tip above a Cp ring. An isolated Nc yields a similar spectrum.

Interestingly, if we position the tip above the center of the Cp ring, an excitation is detected at ± 28 meV above the layer-integrated Nc, but not above the isolated Nc (Supplementary Fig. S4). We assign this excitation to a vibron-assisted spin excitation. A similar conclusion was recently drawn in a study conducted on Nc adsorbed on a silver surface [PRL **123**,106803 (2019), now cited as Ref. 46] despite the very small inelastic conductance reported in that study, *i.e.* a few percent of the total (inelastic+elastic) conductance against several 100% in the present work.

Our findings show that for the copper surface the Nc layer plays an important role in favoring the emergence of vibrational excitations. In previous work (Ref. 29), we showed that the bond of a layer-integrated Nc to a copper surface is weaker compared to an isolated Nc. The weaker bond for a layer-integrated Nc probably favors longer excitation lifetimes, hence stronger vibrational steps in the dI/dV.

Please note that the reference to Fig. S5 in the caption of Fig. 2 was a typo and has been removed.

We have modified the manuscript text on lines 183-189, corrected the caption of Fig. 2 and added Fig. S4. We also added Ref. 46.

(3) I am wondering if the authors have some ideas on why the other two vibrational modes (29.1 and 31.8 meV) as predicted by DFT are not detected from the point of view of selection rules.

Inelastic electron tunneling spectroscopy (IETS) usually involves many electronic states of the molecule and of the substrate. Approximate selection rules can be obtained assuming that the tip does not break the symmetry of the adsorbed system, that the molecular character prevails and that the electron-vibration coupling does not mix states at very different energies [see our PRL **86**, 2593 (2001) and PRL **100**, 226604 (2008)]. In the present case, the molecular symmetry is low enough to be absent from the electron-vibration matrix elements. Given this low symmetry, selection rules cannot be derived. We may only remark that the lower symmetry of the layer-integrated NiNc leads to the lifting of the degeneracy of the 31.8 and 35.5 meV modes, while these modes are degenerate in the isolated NiNc system thanks to its higher symmetry.

(4) The excitations 2 and 3 may come from the spin excitations of two coupled Ni atoms in the NiNc complex. How to rule out this possibility?

This possibility can be ruled out remarking that the Ni atom has a magnetic moment close to zero ($0.2 \mu_B$). Nc carries the magnetism of the NiNc complex as we state in the manuscript (line 90). To complete our answer, we should stress that the magnetism of these systems cannot be reduced to that of the Ni atoms. The cyclopentadienyl rings carry half of the spin density of Nc; half of the spin is on the two rings, the other half by the Ni atom. Modeling the NiNc complex by two interacting Ni atoms would then omit the role played by the rings and oversimplify the problem.

(5) Have the authors studied the tip-height dependence of the dI/dV spectra of layer-integrated NiNc? Since the tip would affect the molecular vibrations, a shift of the vibrational energy or the change of the inelastic conductance is expected. These observations should be helpful to confirm the vibrational origin of the dI/dV features.

We thank the Reviewer for this remark. The energy onset of a vibrational excitation is known to shift due to tip-induced changes in the molecule hybridization or to a Stark shift (Refs. 38 and 39). Following the Reviewer's suggestion, we have studied the tip-height dependence of the dI/dV spectra of a layer-integrated NiNc. The tip was positioned in the center of the NiNc to maximize the signal of the vibrational excitation. The current was then varied from 50 pA to 1 nA. As we show in Fig. 2c, we observed a 1.5 meV red shift of peak 2 and 3, while the position of peak 1 remained nearly unchanged. This further confirms the involvement of a vibrational excitation in peaks 2 and 3.

We have modified the manuscript text on lines 104-106, and added Refs. 38 and 39.

(6) To distinguish the spin and vibrational features in the dI/dV, the authors may want to measure the dI/dV with a nickelocene-terminated tip. Only the spin excitation energy should split or shift with the presence of a magnetic tip. This is just an option.

We have investigated NiNc with a Nc-tip as suggested by the Reviewer (see figure below). The tip height and tip lateral position above NiNc was varied. Panel (a) sketches the experimental setup and panel (b) presents the dI/dV spectra (blue curve: spectrum acquired above the center of NiNc; green curve: spectrum acquired above the Cp ring of NiNc). As shown, the spectra exhibit all the spectral features found when using a metallic tip. The amplitude of the vibrational excitation and of the vibron-assisted excitation vary laterally above the NiNc complex. The behavior is similar to that observed with a metallic tip [Fig. 2(g)]. We decreased the tip-NiNc distance to evidence an exchange coupling

[panels (c) and (d)]. The tip was positioned above the center of NiNc as the off-center position was too unstable. We could not detect an exchange coupling on the low-energy peaks and dips, which are due to single and double spin excitations. The exchange coupling is too weak to be detected. This result is consistent with previous work where we used the Nc-tip above Nc (Ref. 8). It is surprising in view of the work of Czap *et al.* [Science **364**, 670 (2019)] and of our own work [Science **366**, 623 (2019)], and definitely deserves further attention in future work. The spectra acquired with the Nc-tip do however show that the Nc-tip can quench steps **2** and **3** when sufficiently close to the NiNc complex [panel (c)]. This behavior nicely confirms that their amplitudes are related to one another. It also shows that the vibron-assisted spin excitation is easier to detect than the vibrational excitation associated to it.

(7) Experimental details (for example, how they cleaned the Cu substrate, dosed molecules and Ni atoms, lock-in parameters) should be moved to the Method section. I found them distracting when reading the paper.

We have created a “Methods” section at the end of the manuscript where we provide experimental and theoretical details.

(8) Ref. 44 (Nano letters, 16, 588) from the same group already reported the method of producing metal-metalocene complex, and thus should be cited when describing how the NiNc complex was built in the current manuscript.

Done

(9) Figs. 3a and 3b. Scale bars and color scales are missing; Fig. 3b. The energy of the density of states should be stated.

A color-scale was added and the DOS energy is now stated in the caption.

Response to Reviewer #2

N. Bachellier et al., have studied NiNc organometallic molecules using STM/STS. The authors have observed spin and vibrational excitation and the sum of the two excitations. Recent studies of Nc and NiNc complexes from the same group are definitely interesting in exploring molecular spin systems. However, I do not think that this work meets the high standard of novelty required by Nature Communications.

We thank the Reviewer for the report. We understand that the novelty of our work was not properly expressed in the previous version of the manuscript. Let us clarify this point here after.

(1) There have been numerous published results on measuring spin, vibrational, vibronic, spin-vibrational, Kondo-vibrational, and Kondo-spin IETS excitations of single atoms and molecules using STM/STS. In particular, there are two important works which already have reported spin and vibrational excitation. For example, Q. Dubout et al (PRL 114, 106807 (2015)) have performed STM measurements on individual Co atoms on Pt(111). Depending on chemical adsorption of H atom to Co atom, Q. Dubout et al., have not only measured spin anisotropy, Kondo, and vibrational IETS but also have manipulated the spin states of the Co atoms by controllably detaching H atom one by one. Furthermore, Q. Dubout et al., have distinguished the spin features vs vibrational features by measuring the shift of the IETS step with applying external magnetic fields. Furthermore, Y.S. Fu et al (PRL 103, 257202 (2009)) have studied multi-layered CoPc molecules on Pb(111) surface using STM. Y.S. Fu et al., have measured spin-IETS as well as vibronic features and have utilized the change of the spin-IETS for identifying a charge state of molecules. I think that previous researches in this field already cover the most of this work done by N. Bachellier. Also, compared to the two papers mentioned above, I found that this work does not display enough controllability of the spins in the content of nanoscale spintronics.

The studies cited by the Reviewer in his/her answer are impressive, but they focus on a different aspect: the spin manipulation through the doping by hydrogen atoms (Dubout *et al.*) or a charge injection (Fu *et al.*). In both papers, IETS is used to unveil spin excitations and vibronic features. But spin control and manipulation is not at all the topic addressed in our manuscript. Here, we unveil a new type of spin excitation that combines spin and vibrational excitations. This excitation was never reported previously, and certainly not by Dubout *et al.* and Fu *et al.*. The combination of different types of excitations into a combined one opens the fascinating perspective of pumping energy into a degree of freedom by acting on the other, and could even permit access to hidden modes. These aspects are clearly emphasized in the modified version of the manuscript. In view of these modifications, we hope that the Reviewer will reconsider the lack of novelty of our work.

The abstract and the introduction have been modified to make it clear that the main topic of the manuscript is not spin manipulation and/or spin “control”. We added **Refs. 21 and 22**.

(2) The authors claim that a molecule shows the IETS step at the sum of the two excitations (spin-IETS and vibrational IETS). However, I found that the authors’ claim is not convincing to me. For example, what if the IETS step of #2 is spin feature localized at the center of the complex? Could the authors perform magnetic field dependence measurement? Even though the authors’ claim is correct, there have been already many published results showing vibration assisted spin (Kondo) excitations. For example, T. Choi et al (Nano Lett 10, 4175 (2010)) have shown vibration assisted Kondo excitations which can be switched on/off reversibly by controlling the conformation of molecule.

Question (2) has multiple points to be addressed:

Point 1: As we show now in **Fig. 2c**, step #2 exhibits a meV-shift in energy when the tip approaches NiNc contrary to step #1. This tip-induced shift is typical of a vibrational excitation.

Another argument against step#2 being a spin excitation is that there is no spin-Hamiltonian that would possibly describe our IETS data. We see two possible spin-Hamiltonians, both leading to unphysical solutions:

- The first one includes an in-plane magnetic anisotropy E . If the spin of NiNc were an integer, for example a spin $S=1$, one spin excitation would fall at $D-E$ and the other at $D+E$ (see Ref. 6), yielding unrealistic values of the magnetic anisotropy of $D=19$ meV and $E=15$ meV. If the spin of NiNc were half integer ($S=3/2$), we would have only one spin excitation. The half-integer value of the complex could eventually promote a Kondo effect, hence a resonance near zero bias.

- The second one includes a JS_1S_2 coupling between the two Ni atoms of the complex. This possibility can be ruled out remarking that the Ni atom beneath Nc has a magnetic moment close to zero ($0.2 \mu_B$). Nickelocene carries the magnetism of the complex as we state in the manuscript (line 90).

So, a bare spin excitation cannot explain the observation of step #2, and a more complex mechanism needs to be invoked

Point 2: We performed measurements in a magnetic field (≤ 2.7 Tesla) at a temperature of 1.2 K (JT-STM by SPECS GmbH). The data is shown here below. The results could not evidence a clear spin splitting in the line shape as one would expect from the lifting of the two-fold degeneracy of the $M=\pm 1$ states of NiNc. However, this is not surprising as higher fields (>4 Tesla) are necessary to evidence magnetic effects in Nc as recently demonstrated by Czap *et al.* [Science **364**, 670 (2019)] and by our own work [Science **366**, 623 (2019)]. Please keep in mind that the mere presence of a cusp above the excitation energy of steps 1 and 3, which is associated to Kondo-like phenomena, is sufficient to prove that these steps are spin related.

Point 3: Reviewer believes that our work is not sufficiently new in view of existing publications on the vibrational Kondo effect (Refs. 23-28 in the manuscript; in the new version of the manuscript, we have added the work by T. Choi *et al.* as Ref. 26). We disagree with the Reviewer. The vibrational Kondo effect is limited to degenerate spin-systems, and does not correspond to a “spin excitation” as it does in our case. We show for the first time that a similar excitation mechanism can be present in a system with higher spin and, moreover, possessing magnetic anisotropy —contrary to the Kondo effect, the spin-flip scattering is inelastic here. **Hence, we demonstrate the general character of this mechanism and its potential presence in all magnetic systems.**

(3) The authors have assumed the complex has a uniaxial anisotropy as shown in eq. 1. However, the authors also have mentioned the complex has a canted geometry from Fig. 1a and Fig 2c. I believe the canted geometry may introduce in-plane magnetic anisotropy. So, I think the authors need to include in-plane anisotropy in eq. 1 and calculate the IETS energies as well as amplitude of the tunneling conductance at those IETS steps (C.F. Hirjibehedin et al., Science 319, 1066 (2007)).

The uniaxial anisotropy is fully justified from the shape of the first peak (dip), since a finite amount of transversal anisotropy would appear as a split peak (dip) [see point 1 in answer **(2)**]. Moreover, the line shape of the spectrum is similar to that of nickelocene in the layer, where a uniaxial anisotropy is found. Hence, we can safely discard the existence of in-plane anisotropy. This is not surprising as the anisotropy of Nc is imposed by the molecular structure, unlike the Fe atom studied by C.F. Hirjibehedin *et al.* where the anisotropy is induced by the adsorption on Cu₂N.

The Hamiltonian [Eq. 1] used in the manuscript is therefore fully justified. As we explicitly state now in the manuscript (line 96), the quantization axis is chosen along the molecular axis running through the center of the cyclopentadienyl rings. Let us stress that the Hamiltonian of Eq. 1 is also used for describing Nc-terminated tips where the Nc geometry is canted [Ref. 8, Science 364, 670 (2019), and Science 366, 623 (2019)].

(4) In Fig. 3e, the authors have modeled that the IETS #3 is from vibrational mode (n=1) assisted spin-IETS. Have you observed the excitation at the energies of n being higher than 1 (ex, n=2,3,...) (For example, vibronic: X.H. Qiu et al., PRL 92, 206102 (2004), N.A. Pradhan et al., J. Phys. Chem. B 109, 8513 (2005), and spin+vibronic: Y.S. Fu et al (PRL 103, 257202 (2009)). In addition, why does the dI/dV map imaged at the IETS #3 (figure 2e) only show the localized signal at the center of complex? As the authors mentioned, the main contribution of IETS #3 is originated from magnetic anisotropy (IETS #1). However, the signal of the IETS #1 is localized at the ring of the complex not the center of the complex. Could the authors explain this discrepancy?

This is an interesting point. The Reviewer asks whether we observed higher order satellite peaks at higher harmonics of the vibrational mode (n=2, 3 etc.), as other studies found for the observation of vibronic peaks. We did not find features at higher values, but this is quite usual for existing literature reporting the vibrational excitations in the electronic ground state.

The Reviewer would like us to explain why excitation #3, which has magnetic origin, shows a different spatial distribution than #1 where the strongest magnetic effect is located on the cyclopentadienyl rings. We note that as a combined vibrational and spin excitation, the amplitude of #3 corresponds to the product of distributions of the vibrational and spin states. This aspect is of importance and has been addressed in the manuscript on lines 152-159. Since the vibrational mode is more localized, its shape dominates in the maps.

Response to Reviewer #3

Bachellier and co-workers investigate complexes of a Ni atom with a nickelocene molecule. The authors find that in all cases the Ni atom sits below one of the Cp rings of the nickelocene molecule and provides a bond to the underlying copper surface. Interestingly, the authors find two very different adsorption geometries which lead to drastically different spectroscopic signatures in tunneling spectroscopy. First, when the complex is isolated from neighbors, the authors observe a Kondo effect with a surprisingly high Kondo temperature of 68K. Second, when the nickelocene complex is part of a network of molecules, the bonding geometry shows a lower symmetry due to van-der-Waals bonding to neighbors. This results in a visible tilt of the molecule and to a complete absence of the Kondo effect. Rather the spectra indicate a spin excitation, a vibrational excitation, as well as the combination of the two: a spin-vibration excitation at the sum of the two energies. Many results are convincingly confirmed and illustrated through spin-polarized density functional theory modeling.

This work is original, the manuscript is well written and the content of the spin properties of this class of molecule is a hot research topic at present. I am strongly in favor of publishing with minor modifications.

We thank the reviewer for the very positive view of our work. The reviewer finds the manuscript suitable for publication, but has technical comments that we carefully address here below.

(1) Page 1, line 42: I would suggest the authors distinguish the NiNc complex in the Nc network from the stand-alone NiNc because the vibrational mode is induced in the Nc network, is my understanding correct? The sentence can lead to misunderstandings as if the sizable vibrational mode occurred in NiNc itself.

The sentence was changed (line 44).

(2) Page 1, line 44 “previous excitations” is neither defined nor clear from context.

We have rephrased the sentence (lines 44-48).

(3) In Figure 2(c) and page 2, line 62, the apparent height of Nc is given as 4.1\AA , while it is given as 3.5\AA in page 5, line 149. Does the isolated molecule have a different apparent height compared to the one in the Nc layer? In Figure 2(c), the layer (green) and the isolated molecule (blue) look almost the same. If there is tip-dependence, the error given as 0.1\AA seems too low.

Layer-integrated Ncs have a higher apparent height compared to isolated Ncs due to Van der Waals interactions among molecules in the layer (Ref. 29). The “horizontal” layout of Fig. 2(c) smears out the difference between the two apparent heights. The difference is more marked in Figure 2(d) of Ref. 29. We concur with the Reviewer that an error of 0.2\AA is more appropriate. We have set the error bar to 0.2\AA throughout the manuscript.

(4) In Figure 2, image scale for (c-e) is missing. Same for Figure 3 (a,b).

We added a color-scale on both figures.

(5) In Figure 2(a), the authors measured the two spectra at a different set point. Is there a special reason for this? If it is on purpose (to keep the sample-molecule distance same or other reason?), it would be helpful to mention it.

No, there is no particular reason. The spectra do not show a tip-height dependence within the narrow tip-excursion range explored in Fig. 2a.

(6) Page 3, line 85: the magnetic moment of the Ni atom and the nickelocene molecule are not close to

integer or half-integer values. Yet the authors state without comment that the total spin is $S=1$. This seems to fall from the sky and should be discussed with more care.

In order to introduce the actual spin Hamiltonian that describes the system, on Page 3, we have added a small description comparing the spectra of well-known $S=1$ NiNc molecules with the present spectra, and we find an excellent match.

(7) Page 3, line 90: the value D is different from ϵ_1 . What is the relationship between the two values? If it is not trivial, it would be helpful for readers to mention or cite it.

The magnetic anisotropy D is estimated from our fit based on the dynamical scattering model of Ref. 34. The value of D refers to the fit of Fig. 1a. The step position ϵ_1 corresponds to an average value as determined using the data of various NiNc molecules and tips. This was clarified in the manuscript (lines 81, 98-99).

(8) Page 4, line 108: the sentence ‘Second, 3 bears spectroscopic...’ is not clear and easy to understand. Could the authors clarify this sentence?

We have rephrased the paragraph (lines 121-127).

(9) Page 4, line 111: $dI_2/dV^2 \diamond d^2I/dV^2$

Corrected

(10) Figure caption of figure 4: I would state the FWHM of the curve – or the Kondo temperature, to help the reader. In Fig 4(c), the x-axis may be in nm. And, the caption for (d) is labeled as (c).

Done

Reviewers' Comments:

Reviewer #1:

Remarks to the Author:

The authors thoroughly considered the points raised by the referees. Especially, I am satisfied that the authors carried out the suggested experiment and confirmed the vibrational origin of the excitations 2 and 3. Thus, I recommend this work for publication in Nature Communications.

Reviewer #2:

Remarks to the Author:

I have read through carefully the revised manuscript from N. Bachellier et al. The authors have shown additional figures (ex, Fig. 2(c) and Fig. 3(a)) and revised the manuscript according to my comments. I believe that the manuscript shows better in terms of novelty. I have only a few minor comments.

1) In the line of 105-106: The authors mentioned that "We observe a red shift of the peak associated to 2 as high as 1.5meV ...". Could the authors see the same amount of shift in the peak 3 as well? It will be nice if the authors show that the amount of shifts in the peak 1, 2, 3 as a function of set current as a table (maybe in supplementary material)

2) Also, the authors mentioned in the response letter that the tip-induced shift is typical of a vibration excitation. However, this is not necessarily true. As the tip becomes closer to molecules, the electric field at the junction becomes stronger and in turn, this electric field can modulate the charge distribution around molecules, modifying local magnetic anisotropy of the system and shifting spin-IETS features. However, I agree with the authors that peak 1 didn't shift while the peak 2 and 3 did.

Therefore, I could recommend this work to be published in Nature Communication with the minor modification.

Reviewer #3:

Remarks to the Author:

The authors have properly addressed my comments. I am good to publish.

However, I can see that reviewer 2 had stronger reservations - some of which I can follow, others seem overblown to me. I do not fully follow this reviewer and assume that you have to wait for his/her response....

We are pleased to resubmit a revised version of our manuscript. We thank the reviewers for their comments and hope that the modifications meet their expectations. Changes to the manuscript are highlighted in red. Please note that we have edited the manuscript to comply with the format requirements of Nature Communications.

Reviewer #1 and #3 believe that we have properly addressed their comments and both recommend publication in Nature Communications. Reviewer #2 is also satisfied with the new version of the manuscript, as it “shows better in terms of novelty”. He/she favors publication, provided that two minor concerns are properly addressed. We address these issues here after:

1) In the line of 105-106: The authors mentioned that “We observe a red shift of the peak associated to 2 as high as 1.5meV ...”. Could the authors see the same amount of shift in the peak 3 as well? It will be nice if the authors show that the amount of shifts in the peak 1, 2, 3 as a function of set current as a table (maybe in supplementary material)

As suggested by Reviewer #2, we have added a table in the supplementary material (Supplementary Tab. 1).

2) Also, the authors mentioned in the response letter that the tip-induced shift is typical of a vibration excitation. However, this is not necessarily true. As the tip becomes closer to molecules, the electric field at the junction becomes stronger and in turn, this electric field can modulate the charge distribution around molecules, modifying local magnetic anisotropy of the system and shifting spin-IETS features. However, I agree with the authors that peak 1 didn't shift while the peak 2 and 3 did.

We have added the following remark in the caption of Supplementary Tab. 1: “*The position of peak (dip) 1 is nearly constant with tunnel current, i.e. tip-molecule distance. If present, changes in the magnetic anisotropy of NiNc due to the electric field are therefore below detectability [2, 3].*”

Two new references were added to the Supplementary Material (Refs. 2 and 3).